# Fli-1 Activation through Targeted Promoter Activity Regulation Using a Novel 3’, 5’-diprenylated Chalcone Inhibits Growth and Metastasis of Prostate Cancer Cells

**DOI:** 10.3390/ijms21062216

**Published:** 2020-03-23

**Authors:** Youfen Ma, Bixue Xu, Jia Yu, Lirong Huang, Xiaoping Zeng, Xiangchun Shen, Chunyan Ren, Yaacov Ben-David, Heng Luo

**Affiliations:** 1State key laboratory of functions and applications of medicinal plants, Guizhou medical university, Guiyang 550014, China; mayoufen2020317@sina.com (Y.M.); bixue.xu@gzcnp.cn (B.X.); yujia@gzcnp.cn (J.Y.); xiaoping.zeng@gzcnp.cn (X.Z.); shenxiangchun@126.com (X.S.); 2The Key Laboratory of Chemistry for Natural Products of Guizhou Province and Chinese Academy of Science, Guiyang 550014, China; 3Boston Children’s Hospital, Harvard Medical School, Boston, MA 02115, USA; chunyanren@gmail.com; 4College of Food and Pharmaceutical Engineering, Guizhou Institute of Technology, Guiyang 550003, China; hlrong126@126.com; 5College of pharmacy, Guizhou Medical University, Guiyang 550029, China

**Keywords:** prostate cancer, Fli-1 agonist, chalcone, invasion, migration, apoptosis

## Abstract

The friend leukemia integration 1 (Fli-1) gene is involved in the expression control of key genes in multiple pathogenic/physiological processes, including cell growth, differentiation, and apoptosis; this implies that Fli-1 is a strong candidate for drug development. In our previous study, a 3′,5′-diprenylated chalcone, (E)-1-(2-hydroxy-4-methoxy-3,5-diprenyl) phenyl-3-(3-pyridinyl)-propene-1-one (**C10**), was identified as a novel anti-prostate cancer (PCa) agent. Here, we investigated the molecular mechanisms underlying the anti-cancer effects of **C10** on the growth, metastasis, and invasion of PC3 cells in vitro. Our results show that **C10** exhibited a strong inhibitory effect on proliferation and metastasis of PC3 cells via several cellular and flow cytometric analyses. Further mechanism studies revealed that **C10** likely serves as an Fli-1 agonist for regulating the expression of Fli-1 target genes including phosphatidylinositol 3-kinase (*P110*), murine double minute2 (*MDM2*), B-cell lymphoma-2 (*Bcl-2*), Src homology-2 domain-containing inositol 5-phosphatase 1 (*SHIP-1*), and globin transcription factor-1 (*Gata-1*) as well as the phosphorylation of extracellular-regulated protein kinases 1 (*ERK1*). Further, we confirmed that C10 can regulate the expressions of vascular endothelial growth factor 1 (*VEGF-1*), transforming growth factor-β2 (*TGF-β2*), intercellular cell adhesion molecule-1 (*ICAM-1*), p53, and matrix metalloproteinase 1 (*MMP-1*) genes associated with tumor apoptosis, migration, and invasion. Thus, **C10** exhibits stronger anticancer activity with novel molecular targets and regulatory molecular mechanisms, indicating its great potency for development as a novel targeted anticancer drug.

## 1. Introduction

Prostate cancer (PCa) is the most common cause of cancer-related male deaths in the United States, with more than 250,000 new cases reported each year and more than 27,000 deaths annually [1]. In China, PCa presents a regional disparity with a higher incidence rate in developed areas and a lower survival rate in rural areas. Several studies have revealed that the actual numbers of patients and deaths are even larger than those reported. Despite the intensive efforts to improve early detection and treatment, PCa, particularly metastatic PCa, remains to be a significant health problem. Surgery, chemotherapy, radiation therapy, and immunotherapy are the major treatment strategies for PCa treatment [2]. The best strategy for cancer treatment is to kill the malignant cancer cells without damaging normal cells; however, current cancer treatment modalities are far from achieving such a goal because of the limited efficacy of treatment measures on malignant cells and the side effects resulting from the toxicity of drugs on normal cells or immune responses [3,4,5]. Therefore, the development of novel targeted drugs with higher selectivity and lower toxicity is of the utmost importance. 

Based on the history of drug development, natural product-derived compounds have been demonstrated to have unique capabilities for inducing apoptosis more commonly in cancer cells than in normal cells [5,6,7]. This is likely because their unique structures are strictly associated with their binding ability to specific molecules involved in particular biological and/or metabolic pathways [8]. In recent years, statistical studies have also shown that natural compounds are more likely to be developed into new drug candidates than chemically synthesized compounds, with approximately 40% of drugs directly or indirectly being developed from natural products [9]. In fact, several conventional chemotherapeutic agents including Taxol, epothilones, and vinca alkaloids are derived from natural products [5,10]. Nowadays, many studies have focused on the crosslink between natural products or their derivatives and cancer cell apoptosis, clarifying the underlying molecular mechanisms, such as inducing adaptive and apoptotic pathways for inhibiting tumor growth and modulation of ER stress-associated pathways to decrease resistance to chemotherapeutic agents [5,11,12,13]. 

Several studies have shown that Friend leukemia virus integration 1 (Fli-1), a transcription factor in the E26 transformation-specific factor (ETS) family, plays an important regulatory role in the generation of vascular endothelial cells and proliferation of tumor cells [14]. Fli-1 is a DNA-binding protein involved in the expression of several key factors in physiological processes including cell proliferation, differentiation, and apoptosis [15]. Normally, Fli-1 is expressed only in vascular endothelial cells and hematopoietic stem cells. However, recent studies have reported that Fli-1 is also abnormally expressed in various malignant tumor tissues, such as breast cancer, small-cell lung cancer, and bladder cancer cells. Thus, Fli-1 not only plays an important role in blood cells and angiogenesis but also regulates carcinogenesis and the development of various tumors [16]. Therefore, Fli-1 as a specific target for anti-cancer drugs can regulate the proliferation, differentiation, and apoptosis of tumor cells and block the formation of tumor blood vessels for controlling cancer cell invasion and migration.

Chalcone is an important group of flavones containing two phenyl rings (A- and B-rings) connected by a three-carbon α, β-unsaturated carbonyl bridge [17]. Given this special structure, prenylated chalcone has various pharmacological properties including anti-bacterial, anti-inflammation, and anti-cancer properties [18,19,20]. We previously designed and synthesized a series of novel 3′,5′-diprenylated chalcones aiming to identify those having an anti-proliferative effect. We identified three compounds showing inhibitory activity on tumor cell proliferation in vitro by inducing cell apoptosis and cell cycle arrest, and (E)-1-(2-hydroxy-4-methoxy-3,5-diprenyl) phenyl-3-(3-pyridinyl)- propene-1-one (**C10**) exhibited potent anti-cancer activity with better selectivity and lower toxicity than the other two in leukemia cells [18,21]. Our research group conducted a systematic study on anticancer drugs targeting Fli-1. After analyzing the functional sequence of Fli-1 gene, we developed a strategy for screening compounds that can regulate Fli-1 expression based on promoter activity [22]. Based on this strategy, we hypothesized that C10 might act as an agonist for Fli-1 expression in prostate cancer cells, which was opposite to the role of Fli-1 in other cancers. For example, professor Ben-David’s group has found that Fli-1 was a proto-oncogene in leukemia, and inhibiting the expression of Fli-1 could inhibit the proliferation of leukemia cells and induce apoptosis [23]. Therefore, interestingly, we focused on the opposite conclusion and want to investigate the molecular mechanism. However, the effect of **C10** on the growth, invasion, and migration of prostate cancer cells and the molecular mechanisms of **C10** has not yet been elucidated. In this study, we aimed to explore the detailed molecular mechanism of **C10** to determine its anti-cancer activity toward human prostate cancer cells.

## 2. Results

### 2.1. C10 Promoted the Fli-1 Expression in PC3 Cells

We have previously established a strategy to screen the target modulators of Fli-1 from the natural products library of traditional Chinese medicine using a luciferase-based expression assay from the readout of FB-Luc, a reporter plasmid in which two consensus Fli-1-binding sites are inserted upstream of a minimal promoter [24]. On screening **C10** for promoting or repressing effects on FB-Luc luciferase activity via high-content screening in 96-well plates (Figure 1), **C10** was shown to significantly activate Fli-1 promoter-dependent luciferase expression in the presence of Fli-1, suggesting its role as an Fli-1 agonist for activating Fli-1 minimal promoter (Figure 1A). When co-transfected with MigR1-Fli-1 expression vector in HEK293T cells, **C10** increased the luciferase activity of the Fli-1 reporter plasmid and control plasmid driven by the CMV promoter (CMV-Luc) by approximately 8- and 4-fold, respectively. This implies that **C10** can directly affect Fli-1, independent of Fli-1 minimal promoter to achieve transcriptional activation (Figure 1B). **C10**-induced expression of Fli-1 in PC3 cells was determined via RT-PCR and Western blotting (Figure 1C,D). The results show that Fli-1 expression was significantly up-regulated (*p* < 0.05) in cells treated with 2, 4, and 8 μmol/L of **C10** compared with control cells at both RNA and protein levels. To confirm **C10** and Fli-1 binding in vivo, we performed a cellular thermal shift assay [25] and showed that **C10** binding could stabilize Fli-1 in a concentration-dependent manner without affecting the stability of a non-related protein [glyceraldehyde-3-phosphate dehydrogenase (GAPDH)], indicating that **C10** can bind Fli-1 in cells (Figure 1D). 

The cells were treated with 2 μmol/L of **C10** for 24 h to up-regulate Fli-1 expression, and then Fli-1 expression was knocked down with siRNA, and randomly shuffled sequences of siRNA and were used as negative control (NC, Figure 2). The research strategy is shown in Figure 2A. The results show that the designed siRNA of Fli-1 (40 and 60 nmol/L) effectively reduced Fli-1 expression induced by **C10** in PC3 cells (*p* < 0.01) compared with NC (Figure 2B); **C10**-treated cells were treated with siRNA and NC for 6 h and 42 h to investigate the cell growth inhibition rate, respectively (Figure 2C). The results show that the cell growth inhibition rate of siRNA-treated cells was significantly (*p* < 0.01) lower than that of NC and **C10**-treated cells (*p* < 0.01), indicating that **C10**-induced Fli-1 expression can significantly inhibit cell growth, and reduction in **C10**-induced Fli-1 expression levels can significantly (*p* < 0.01) recover the cell growth ability. These results indicate that Fli-1 is a key binding target of **C10** for inhibiting the growth of PC3 cells.

Next, we used unbiased blind docking to predict the binding region between **C10** and Fli-1 protein using all known DNA binding domain structures of Fli-1 identified by X-ray crystallography (chain A with PDB code 5E8G, 5E8I, and 5JVT) (Figure 3). We found that the docking of **C10** and 5JVT chain A showed the lowest deltaG (−7.98 kJ/mol) and fullfitness score (−730.49). The docking analysis also revealed that **C10** may interact with the binding pockets formed among N- and C-terminal helixes and the loop regions opposite the DNA binding pocket (Figure 3A,B). Residues in the binding sites of **C10** in Fli-1 protein include Leu288, Trp302, Tln305, Asn306, Gly307, Lys359, Phe360, Asp361, Phe 362, and Ile365 (Figure 3C). These results suggest that C10 could directly bind with Fli-1 protein for complex formation. 

### 2.2. C10 Inhibited the Proliferation of PC3 Cells by Inducing Apoptosis

We conducted a preliminary study of the growth assay, apoptosis progression of PC3 cells by C10 in our previous published paper [18]. However, this study was a research paper focusing on chemical synthesis of 3’,5’-diprenylated chalcone derivatives and their primary cytotoxicity. The previous study showed only the inhibition of 5 μmol/L of C10 on cell growth, apoptosis, and cell cycle has been tested in the cell viability section, and no concentration gradient treatment was performed. On the basis of this study, the effects of different treatment time and different concentrations of C10 on cell growth were studied in different prostate cancer cells. We determined the effect of **C10** on PC3 and LNcap cell proliferation by MTT assay using different **C10** concentrations and treatment times (Figure 4A). The results show that **C10** inhibited PC3 cell proliferation in a dose- and time-dependent manner, with IC_50_ values at 48 h and 72 h lower than 2 μmol/L. However, the **C10** IC_50_ values for LNcap cells were larger than those for PC3 cells in the same treatment time (Figure 4A). We observed a reduced cell number with increasing **C10** concentration and apoptotic bodies when the cells were treated with 4 μmol/L of **C10**; this suggests that **C10** induced apoptosis in PC3 cells (Figure 4B). Next, we monitored the apoptotic morphology of PC3 cells via Hoechest 33258 staining (Figure 4C). We observed that chromatin condensation and fragmentation significantly changed in a dose-dependent manner when the cells were treated with **C10** compared with that observed in controls. The apoptosis was further analyzed by flow cytometry (Figure 4D), and we concluded **C10**-induced apoptosis to be late apoptosis based on a quantitative analysis (Figure 4E,F). 

### 2.3. Inhibitory Effect of C10 on Migration and Invasion of PC3 Cells

Changes in the invasion ability of PC3 cells after treatment with different concentrations of **C10** was evaluated via the transwell invasion assay (Matrigel^®^ Matrix in vitro cell membrane-induced cell invasion kit) (Figure 5A). The results show that the number of cells capable of passing through the chamber after treatment with **C10** for 24 h was markedly reduced compared with that after treatment with dimethyl sulfoxide (DMSO) for 24 h (Figure 4A) as quantified (*p* < 0.01) (Figure 5B). These results indicat that the invasion ability of PC3 cells was significantly suppressed by **C10** in a dose-dependent manner.

Changes in the migration ability of PC3 cells after treatment with different concentrations of **C10** was determined by scratch repair experiment, including micrographs obtained from an inverted fluorescence microscope (Figure 5C) and quantitative analysis (Figure 5D). We observed that the cell migration rate was significantly (*p* < 0.01) lower after treatment with different concentrations of **C10** (0, 0.5, 1.0, and 2.0 μmol/L) for 24 and 48 h than after treatment with DMSO for 24 and 48 h. These results suggest that the migration ability of PC3 cells was significantly suppressed by **C10** in a dose- and time-dependent manner.

### 2.4. Regulatory Effect of C10 on the Expression of Fli-1 Target Genes and Metastasis-Associated Genes in PC3 Cells

To explore the molecular mechanism of **C10** in cancer cell proliferation and apoptosis in an Fli-1-dependent manner, we evaluated the expression of Fli-1 target genes (Figure 6) along with apoptosis and metastasis-associated genes (Figure 7) in PC3 cells after treatment with different concentrations of **C10**. Our results indicate that the mRNA levels of Fli-1 target genes (Bcl-2, MDM2, and Gata-1) were down-regulated by **C10** in a dose-dependent manner (2, 4, and 8 μmol/L) compared with DMSO controls as per RT-PCR (Figure 6A,B) and qPCR (Figure 6C). However, the mRNA levels of retinoblastoma (Rb), p110, SHIP-1, and ERK1 did not change. At protein level, p110, MDM2, Bcl-2, and Gata-1 were down-regulated by **C10** in a dose-dependent manner, but SHIP-1 displayed an increasing trend compared with the control (Figure 6D,E). Although the expression of Rb and ERK1 did not change, the level of phosphorylated ERK1 (P-ERK1/2) was observed to decrease with increasing **C10** concentrations. Thus, anti-proliferation activity of **C10** is likely associated with the regulation of Fli-1 target genes along with apoptosis and metastasis-associated genes, including p110, P-ERK1/2, MDM2, Bcl-2, SHIP-1, and Gata-1, and the phosphorylation level of ERK1, which governs cell proliferation and apoptosis (Figure 6F). 

To explore the molecular mechanism of C10 in cancer cell metastasis, we investigated the effect of C10 on metastasis-related factors (Figure 7). We found that common metastasis-related factors such as VEGF-1, ICAM-1, MMP-1, and TGF-β2 were significantly (*p* < 0.01) inhibited by **C10** in a dose-dependent manner at mRNA level, except for matrix metalloproteinase 9 (MMP-9) (Figure 7A–C). A similar inhibition of these metastasis-related factors was observed at a protein level (Figure 7D,E). Notably, the expression level of MMP-9 at a protein level decreased (*p* < 0.01) in cells induced with 8 μmol/L **C10**; the expression level of P53 at a protein level increased (*p* < 0.01) in cells induced with different concentrations of **C10**. We further created a regulatory network for easy visualization of interactions among the abovementioned genes using STRING website (https://string-db.org/). The network showed that the anti-cancer effect of C10 may be attributed to Fli-1 target genes and metastasis-associated genes in two sets: proliferation and apoptosis and migration and invasion (Figure 7F). Based on the abovementioned findings, we proposed a possible molecular pathway for C10 to regulate the occurrence and development of prostate cancer, as described in Figure 8.

## 3. Discussion

In this study, we explored the activity and molecular mechanism underlying the action of **C10**, an Fli-1 agonist, on the growth and metastatic capability of PC3 prostate cancer cells. After multiple experimental and computational approaches, we observed that **C10** exhibited a strong inhibitory effect on PC3 proliferation and metastasis. We showed that **C10** can serve as an Fli-1 agonist to activate the promoter for Fli-1 expression and regulate the expression of Fli-1 target genes, including p110, P-ERK1/2, MDM2, Bcl-2, SHIP-1, and Gata-1, and the phosphorylation of ERK1. **C10** can also regulate the expression of apoptosis-, migration-, and invasion-related genes, including VEGF-1, TGF-β2, ICAM-1, and MMP-1. Our results suggest that the newly synthesized 3′,5′-diprenylated chalcone **C10** is a novel Fli-1 agonist exhibiting strong anticancer activity with new molecular targets and regulatory molecular mechanisms, which supports its great potential to be developed as a novel targeted anticancer drug.

Although we are far from fully understanding the molecular mechanisms of cancer occurrence and progression, many studies have focused on apoptosis, a form of programmed cell death that cancer cells escape from during normal cell division [26]. Indeed, inducing apoptosis has been confirmed as one of the main effective approaches for treating various cancers [26,27]. In terms of drug development, apoptosis induction becomes a main measurement to develop selective and effective anti-cancer agents. Meanwhile, it is crucial to develop agents with better selectivity that only regulate the physiological activity (apoptosis) of cancer cells but not the normal cells. This requirement emphasizes the importance of finding cancer-specific suppressors or oncogenes.

Fli-1 has been reported to play a role in blood, breast, skin, and liver cancer [28,29,30,31], particularly in retrovirus-induced hematologic malignancies [16]. However, its role in epithelial-derived malignancies, particularly prostate cancer, has been less studied. The study of Fli-1 in epithelial-derived adenocarcinoma was limited to breast cancer [32]. Fli-1 expression has been was observed in various breast cancer cell lines, such as MDA, MB231, MB436, BT-549, and HCC1395; further, 31% of patients with mammary medullary carcinoma express Fli-1, suggesting that it is associated with the occurrence of malignant breast tumors. Mechanism studies have shown that the deletion of Fli-1 can promote breast cancer growth metastasis, whereas exogenous expression of Fli-1 can significantly inhibit the breast cancer cells growth, metastasis, and invasion both in cell and mouse models [33]. Furthermore, whole-genome sequencing and RNA sequencing have revealed that over-expression of Fli-1 affects several cell signaling pathways closely associated with cancer, among which WNT, pi3k-akt, and VEGF signaling pathways are significantly affected [34] These studies on breast cancer have shed light upon the up-regulation of Fli-1 expression as a potential strategy to inhibit epithelial-derived tumors. However, the role of Fli-1 in breast cancer remains controversial to date, as several studies have also shown that Fli-1 can inhibit cell apoptosis or promote tumor progression [35].

Based on previous studies, we found that several Fli-1 target genes including Rb, MDM-2, Bcl-2, Gata-1, p110, and SHIP-1 were involved in the regulation of cellular proliferation, apoptosis, and differentiation [31]. Rb, as the earliest discovered tumor suppressor gene, is mainly involved in the regulation of cell cycle and negatively regulates cell growth by processes such as the up-regulation of the expression of proto-oncogene C-myc [35,36]. However, our experimental results showed that **C10** did not regulate Rb expression, but affected the expression of other Fli-1 target genes including p110, MDM2, Bcl-2, SHIP-1, Gata-1, and the phosphorylated form of ERK1. According to previous studies, VEFG-1, ICAM-1, TGF-β2, MMP-1, and MMP-9 are key genes involved in the regulation of cancer cell metastasis and invasion [37,38,39,40]. In this study, we found that **C10** can inhibit the expression of these five key genes at a protein level. Small C10 molecules may first target Fli-1 after entering the cell to promote Fli-1 expression by activating its promoter activity, and then induce apoptosis by regulating the expressions of Gata-1, MDM2, p110, SHIP-1, Bcl-2, and phosphorylated ERK1, control cancer cell proliferation by regulating p110 and SHIP-1, and affect the metastatic ability of cancer cells by regulating VEFG-1, ICAM-1, TGF-β2, MMP 1, and MMP-9.

It should be noted that controlling cancer metastasis is the key to the success of cancer treatment. Approximately 40% of prostate cancer patients relapse after first-line treatment and develop more aggressive and lethal castration-resistant prostate cancer (CRPC) [41]. For treating highly metastatic prostate cancers, therapeutic strategies still mainly rely on targeting the androgen receptor (AR) axis pathway by various means such as inhibiting the synthesis of androgens; targeting the ligand binding region, DNA binding domain, or n-terminal structural domain (NTD) of AR; and using AR antagonists. However, although this approach may extend the survival time to a certain extent, drug resistance will finally lead to the failure of AR signal axis targeting therapy [42]. Therefore, developing new therapeutic targets and corresponding targeted drugs is an urgent requirement. Our study shows that targeting Fli-1 can be such an alternative strategy to control the proliferation and metastasis of prostate cancer. We believe that controlling the metastatic capability of tumors will provide a new strategy and has important clinical significance for the treatment of prostate cancer.

## 4. Materials and Methods

### 4.1. Materials 

All equipment and biological agents used in this study were as previously reported [18,21]. The other reagents were of analytical grade. PC3 and HEK293T cell lines were obtained from the Biology laboratory of the Key Laboratory of Chemistry for Natural Products of Guizhou Province and Chinese Academy of Sciences (Guiyang, China). Cells were cultured in Dulbecco’s Modified Eagle Medium (DMEM) containing 10% fetal bovine serum (FBS) and 1% penicillin and streptomycin (Sijiqing, Hangzhou, China) at 37 °C in a CO_2_ incubator (5% CO_2_ and 95% air, 95% humidity). Eukaryotic expression vector MigR1-Fli-1 was provided by professor Yaacov Ben-David. The authors declare that there is no ethical issue to declare. 

### 4.2. Antibodies

Antibodies against Fli-1 (ab153909, 1:1000), Rb (ab181616, 1:2000), MDM2 (ab38618, 1:1000), Gata-1 (ab181544, 1:1000), Bcl-2 (ab32124, 1:1000), p110 (ab151549, 1:1000), SHIP-1 (ab45142, 1:1000), MMP-1 (ab137332, 1:1000), TGF-β2 (ab113670, 1:1000), phospho-ERK 1/2 (T202 +Y204) (ab223500, 1:1000), and ERK1 (ab32537, 1:1000) were purchased from Abcam (Cambridge, UK). Antibodies against MMP-9 (#13667s, 1:1000), ICAM-1 (#4915s, 1:1000), GAPDH (#2118s, 1:1000), and VEGF-1 (#2893s, 1:1000) were purchased from Cell Signaling Technology (Beverly, MA, USA). Primers (Rb, MDM2, Gata-1, Bcl-2, PI3 K/Ras, SHIP-1, MMP-1, TGF-β2, ERK1, MMP-9, ICAM-1, and VEGF-1) were purchased from GenScript (Nanjing, China). Anti-rabbit and anti-mouse LgG (H+L) [Dylight (TM) 800 4 × PEG Conjugate] secondary antibodies used in this study were purchased from Cell Signaling Technology and used at a 1:30000 dilution in experiments. 

### 4.3. Cell Proliferation Assay

Cell proliferation assay was performed as previously described [18]. Briefly, 6 × 10^3^ cells were seeded and cultured in 96-welled plates to approximately 70% confluency, and then treated with different concentrations of **C10** (0, 2, 4, 6, 8, and 12 μmol/L) for 24 h, 48 h, and 72 h. 3-(4,5-Dimethylthiazol-2-yl)-2,5-diphenyltetrazolium bromide (MTT) solution was then added to each well and cells were further incubated at 37 °C for 4 h. The formazan crystals were dissolved in DMSO and the absorbance was measured at 490 nm using a microplate reader. The cell number and morphology of PC3 cells were determined using a fluorescent inverted microscope at 24 h, 48 h, and 72 h after treating with 4 μmol/L **C10**. The IC_50_ values (μmol/L) of **C10** were determined from inhibition curves at different treatment times. To investigate the effects of Fli-1 knockdown on PC3 cell growth, 5 × 10^3^ cells were continuously treated with 2 μmol/L **C10** for 24 h. Following this, the cells were treated with NC and siRNA for 6 h and replaced with fresh medium and incubated for 48 h. PC3 cell growth inhibition rate in each group was then detected by MTT method.

### 4.4. Scratch Repair Assay

The cell transfer ability was measured via an in vitro scratch repair assay [43]. PC3 cells in logarithmic growth phase were seeded into a 6-well plate at a concentration of 50,000 cells per well and incubated at 37 °C for 24 h. When PC3 cells attached and grew to about 80%, we made scratched lines and added different concentrations (0, 0.5, 1.0, and 2.0μmol/L) of **C10** and incubated for 12 h and 24 h. The repair ability was observed using an inverted fluorescence microscope. The effect of **C10** on the metastasis ability of PC3 cells was analyzed based on the repair ability.

### 4.5. Transwell Chamber Invasion Assay

Transwell chamber invasion assay was used to examine the effect of **C10** on the invasion capability of PC3 cells [43]. First, Matrigel^®^ Matrix (Invitrogen, Carlsbad, CA, USA) was thawed overnight at 4 °C, and diluted in serum-free DMEM to a final concentration of 1 mg/mL. Next, 100 μL of diluted Matrigel was added vertically in the center bottom of the chamber, and incubated at 37 °C for 4–5 h to make Matrigel gelatinous. PC3 cells in the logarithmic growth phase were suspended in serum-free medium, and adjusted to a concentration to 2 × 10^5^/mL. Following this, 600 μL of DMEM containing 10% serum was added to the lower chamber and 100 μL of serum-free cell suspension was added to the upper chamber. After 24-h incubation, the chamber containing PC3 cells was removed and fixed with methanol for 30 min, stained with Giemsa dye solution for 30 min, and rinsed with water several times. The membrane was peeled off with small forceps and allowed to dry. Finally, three random field counts were done using a microscope to analyze the effect of **C10** on the invasion ability of PC3 cells.

### 4.6. Cell Apoptosis Assay

Cell apoptosis level was determined by flow cytometry and Hoechst 33258 assay, as described previously [18,21]. Briefly, 5 × 10^6^ PC3 cells were seeded in a 6-well plate and incubated for 12 h. The cells were treated with different concentrations of **C10** (0, 2, 4, and 8 μmol/L) for 24 h, harvested, and stained with annexin-V-FITC and PI, followed by flow cytometric analysis. Chromatin condensation and fragmentation of apoptotic cells induced by **C10** was visualized under a fluorescent inverted microscope after Hoechst 33258 staining.

### 4.7. Genes Expression Assay

Gene expression was detected at a transcriptional (RT-qPCR and semi-quantitative RT-qPCR) and protein level (Western blotting), as described previously [21]. GAPDH served as an endogenous control. The primer sequences are listed in Table 1. 

### 4.8. Cellular Thermal Shift Assay

The experiment was performed as previously described [44]. Briefly, PC3 cells were treated with an increasing amount of **C10** and DMSO for 1 h at 37 °C. Following this, the cells were washed with PBS, heated at 49 °C for 5 min, placed at room temperature for 3 min, and treated with liquid nitrogen (−37 °C) water-bath cycles three times. Cell lysate was centrifuged at 16,000 *g* for 30 min and the supernatant was collected to analyze the protein expression of Fli-1 and GAPDH by Western blotting. Band intensity was normalized to that of DMSO-treated samples using Image StudioTM Lite (LI-COR, Inc.).

### 4.9. Promoter Activity Assay

The effect of **C10** on Fli-1 promoter activity was assayed as described. Briefly, FB-Luc (1.25 μg) and MigR1 (1.25 μg) or MigR1-Fli-1 (1.25 μg) expression vectors were co-transfected into HEK293T cells using Lipofectamine 2000 (Life Technology, Beijing, China) and incubated for 24 h. The transfected cells were seeded into 96-well plates and incubated for 12 h, followed by treatment with different concentrations of **C10** (0, 2, 4, and 8 μmol/L) for 12 h and subsequent luciferase activity determination, as described previously [45].

### 4.10. Molecular Docking Analysis

The compound **C10** was docked to chain A of the crystal structure (5E8G, 5E8I, and 5JVT) of Fli-1 using SwissDock with default parameters and ranked based on cluster and fullfitness [46]. We used the top-scored predicted binding sites for **C10** to localize the binding region on Fli-1. Protein–ligand interactions were analyzed using LIGPLOT [47]. The parameters used for docking runs: number of BMs (binding mode) generated by the DSS engine: 5000, SD (Steepest descent) minimization steps: 100, ABNR (the adopted basis Newton-Raphson method) minimization steps: 250, number of BMs clustered and evaluated by the FullFitness: 250. The structures were visualized via UCSF Chimera [46,47] and Pymol (PyMOL Molecular Graphics System, Version 1.8 Schrödinger, LLC).

### 4.11. Statistical Analysis

We applied the same statistical analysis methods as previously reported [21]. A student’s test was used for data analysis. Data were analyzed by SPSS 18.0 software and are presented as mean ± SD of three independent experiments. *p* < 0.05 was considered statistically significant.

## Figures and Tables

**Figure 1 ijms-21-02216-f001:**
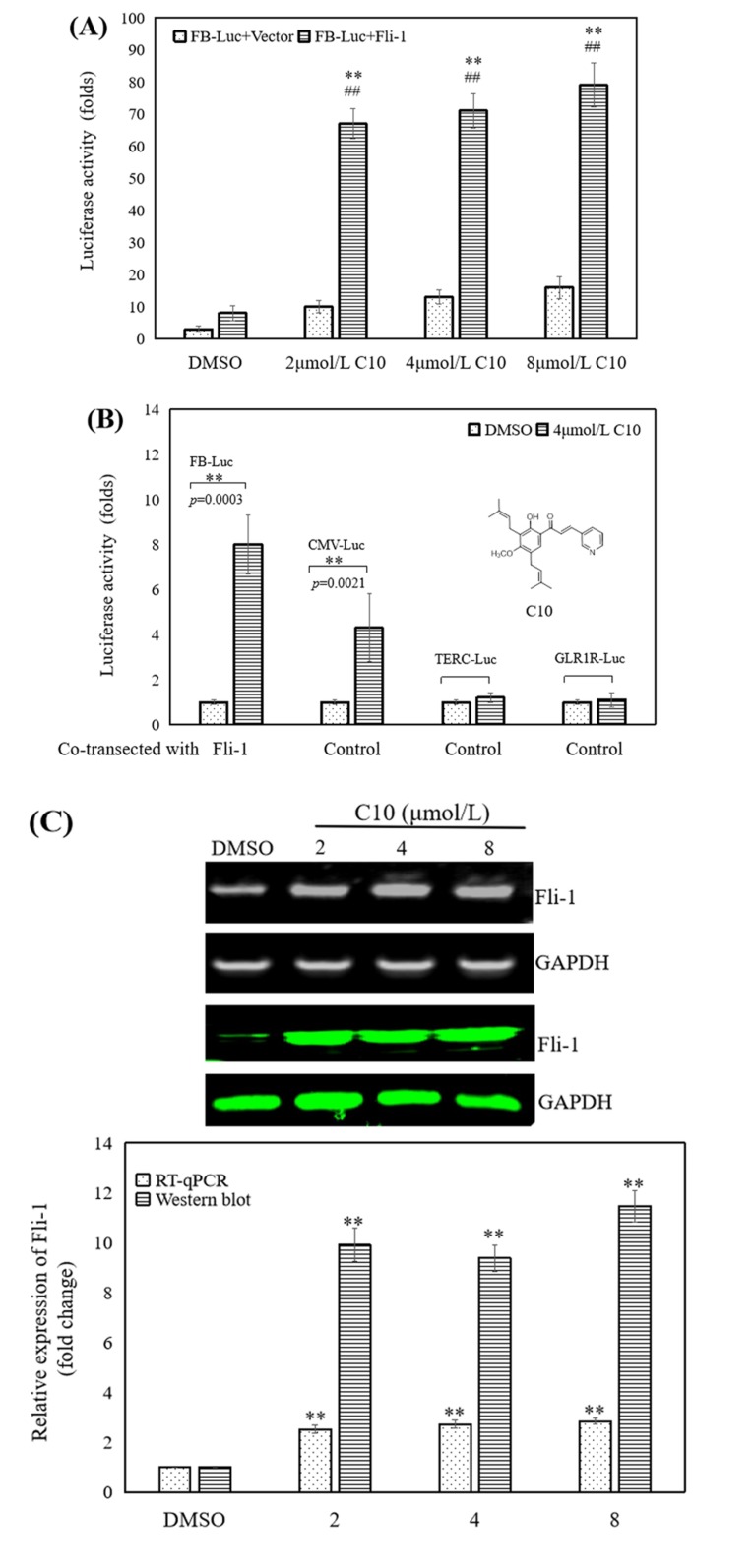
Effect of **C10** on the promoter activity of Fli-1. (**A**) **C10** significantly increased transcriptional activity of FB-Luc (1.25 μg) reporter gene when co-transfected with MigR1-Fli-1 (1.25 μg) or MigR1 (1.25 μg) vectors into HEK293T cells. (**B**) **C10 (**4 μM) moderately increased luciferase activity of control CMV-Luc but not of TERC-Luc or GLP1R-Luc promoters. (**C**) Dose-dependent effect of **C10** on Fli-1 expression in PC3 cells. Fli-1 expression at mRNA and protein levels in PC3 cells exposed to **C10** for 24 h by RT-PCR and western blotting, respectively, ** *p* < 0.01 (*n* = 3) compared with the control by RT-PCR, ## *p* < 0.01 (*n* = 3) compared with the control by RT-PCR. (**D**) Effects of **C10** on protein stability assessed in a cellular thermal shift assay shown as a representative set for Western blot analyses of Fli-1 and GAPDH, ** *p* < 0.01 (*n* = 3) compared with the control expression of Fli-1 treated in 49 °C. ## *p* < 0.01 (*n* = 3) compared with the expression of Fli-1 treated in 37 °C. The histograms show the relative protein expression of Fli-1 in PC3 cells as analyzed using the Image J software. GAPDH was used as a loading control. Data are presented as the means ± SEM from at least three independent experiments.

**Figure 2 ijms-21-02216-f002:**
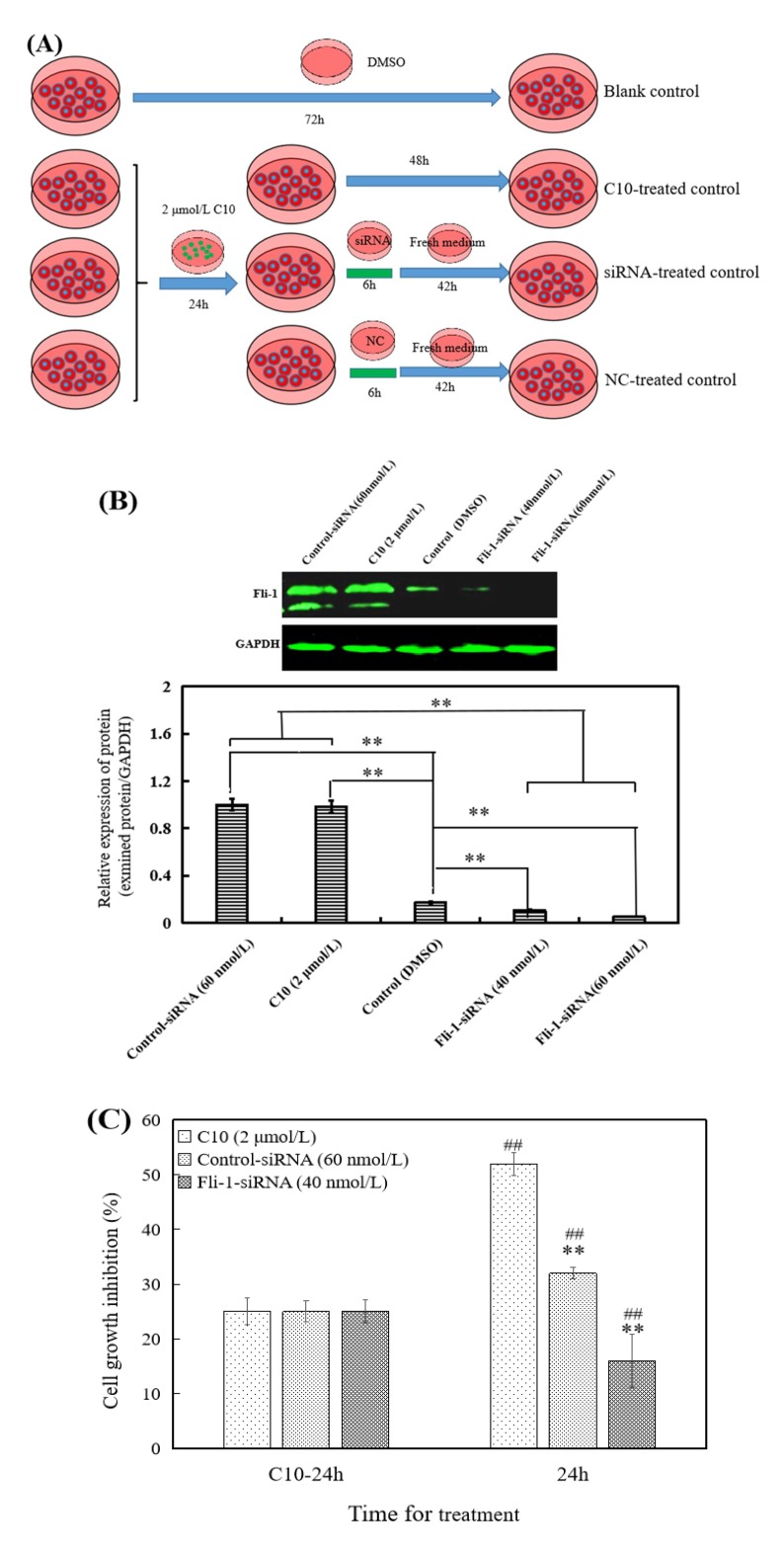
Effects of Fli-1 knockdown with siRNA on cell growth in PC3 cells with **C10-**induced Fli-1 expression. (**A**) The experimental strategy in cell culture and treatment. (**B**) The relative expression of Fli-1 as detected by Western blotting in PC3 cells treated with C10, siRNA, NC, and dimethyl sulfoxide (DMSO) treatment (blank control). ** *p* < 0.01 (*n* = 3) compared with the control. ## *p* < 0.01 (*n* = 3) compared with PC3 treated by C10 (2μmol/L) and control-siRNA. (**C**) Effect of Fli-1 knockdown on cell growth in PC3 cells with **C10-**induced Fli-1 expression via MTT for 48 h. The data are presented as the means ± SEM from at least three independent experiments. ** *p* < 0.01 (*n* = 3) compared with the growth inhibition of the cells with 2 μmol/L **C10** treated for the same time; ## *p* < 0.01 (*n* = 3) compared with the growth inhibition of cells with 2 μmol/L **C10** treated for 72 h.

**Figure 3 ijms-21-02216-f003:**
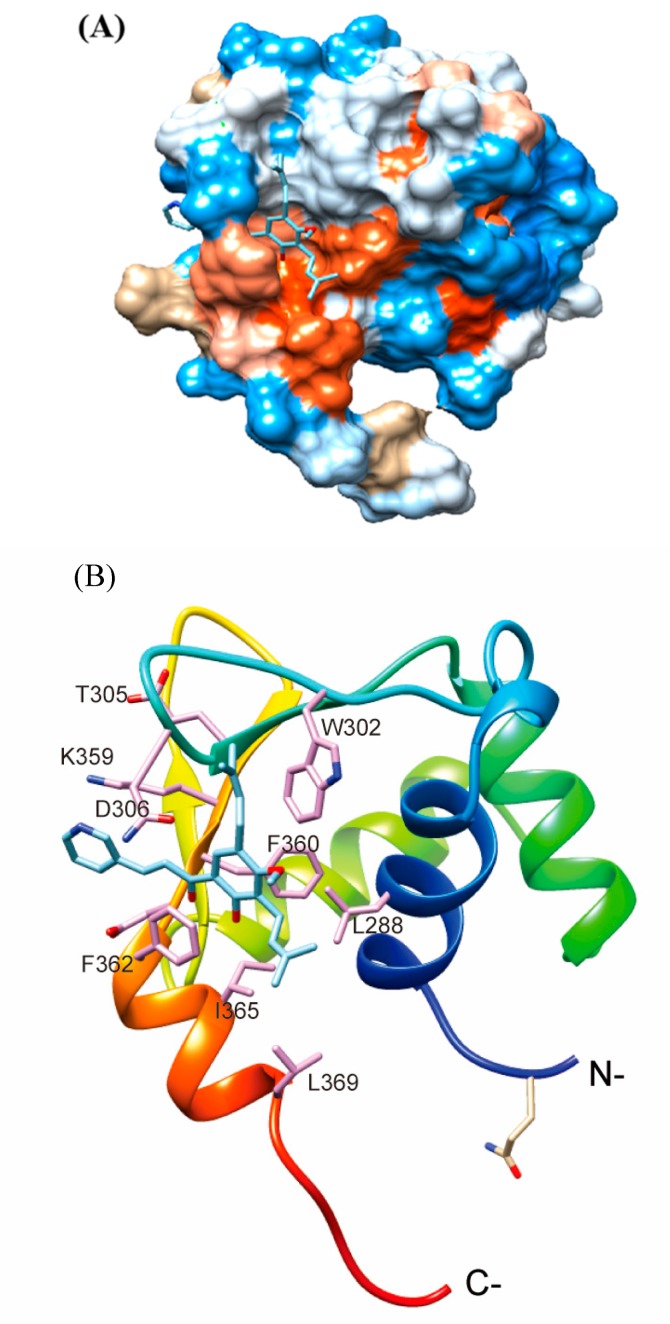
Computational docking analysis of **C10** with Fli-1 protein. (**A**) Surface representation of Fli-1 showing the compound binding pocket on the structured region of Fli-1 DNA-binding domain (PDB: 5JVT, chain A). Red: acidic or negatively charged region; blue: basic or positively charged region; white: neutral region. The predicted binding mode of residues (pink, stick) from Fli-1 DNA-binding domain (rainbow, ribbon) with **C10** (cyan, stick). (**B**) the amino acid binding sites between the Fli-1 protein and **C10**.

**Figure 4 ijms-21-02216-f004:**
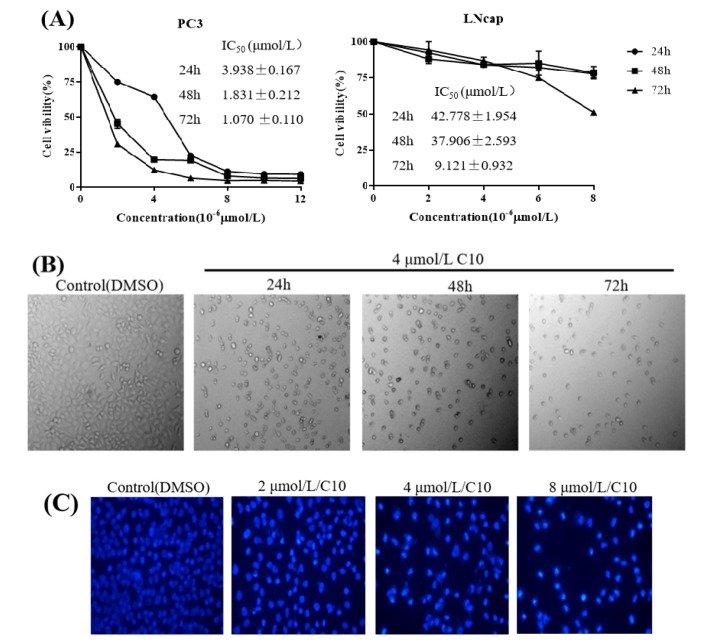
**C10**-induced inhibition of PC3 cell proliferation by inducing apoptosis. (**A**) ell viability curves of **C10** on PC3 and LNcap cell growth by MTT assay with IC_50_ values (μmol/L) for different treatment durations. (**B**) Cell number and morphologic analysis of PC3 cells when treated with 4 μmol/L **C10** using a fluorescent inverted microscope at 24 h, 48 h, and 72 h. **C10-**induced apoptosis visualized by Hoechst 33258 staining (**C**) and flow cytometry (**D**, **E**, and **F**). Chromatin condensation and fragmentation of apoptotic cells induced by **C10** was observed by Hoechst 33258 staining observed using a fluorescent inverted microscope after treatment with different concentrations of **C10** (0, 2, 4, and 8 μmol/L) for 24 h. Harvested cells were stained with annexin-V-FITC and PI, and analyzed by flow cytometry. The statistical analysis for the flow cytometry assay represented the percentage of early and late apoptosis from at least three independent experiments. Data were analyzed using Origin Pro 9.0 and are presented as means ± SEM from at least three independent experiments. ** *p* < 0.01 (*n* = 3) compared with the control.

**Figure 5 ijms-21-02216-f005:**
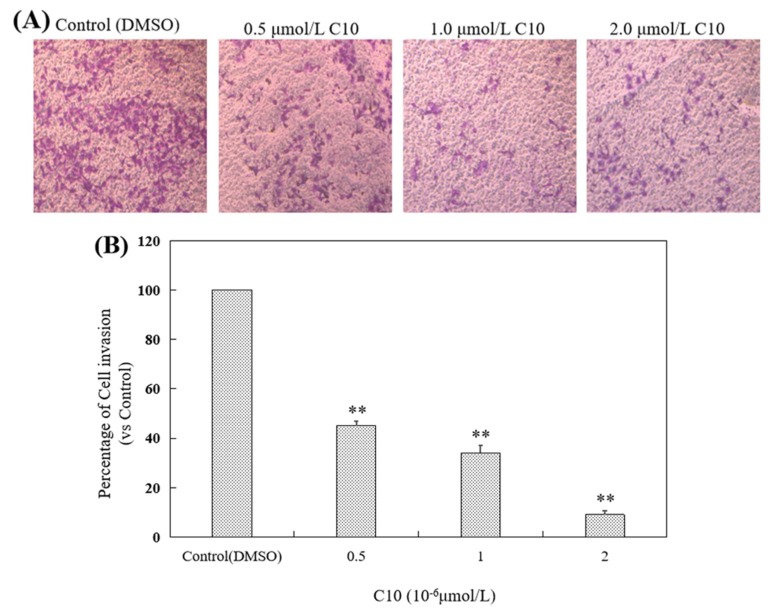
Inhibitory effect of **C10** on migration and invasion of PC3 cells. (**A**) The invasive ability of PC3 cells on treatment with different concentrations of **C10** (0, 0.5, 1, and 2 μmol/L) for 24 h was assayed by transwell invasion assay. PC3 cells were treated with **C10** for 24 h, and stained with 0.1% crystal violet, and observed and photographed via fluorescence microscopy. (**B**) Quantitative analysis of the percentage of invasive cells in one microscopic field. (**C**) Effects of **C10** on the migration ability of PC3 cells. (**A**) The migration ability of PC3 cells on treatment with different concentrations of **C10** (0, 0.5, 1, and 2 μmol/L) for 0, 12, and 24 h was assayed by scratch repair experiment. (**D**) Quantitative analysis of the percentage of migrated cells in one microscopic field. The data are represented as mean ± SD from at least three independent experiments. ** *p* < 0.01, the percentage of migrated cells treated with different concentrations of **C10** compared with that of the control (0 μmol/L); ## *p* < 0.01, the percentage of migrated cells treated with **C10** for different treatment durations compared with that of the control (0 h).

**Figure 6 ijms-21-02216-f006:**
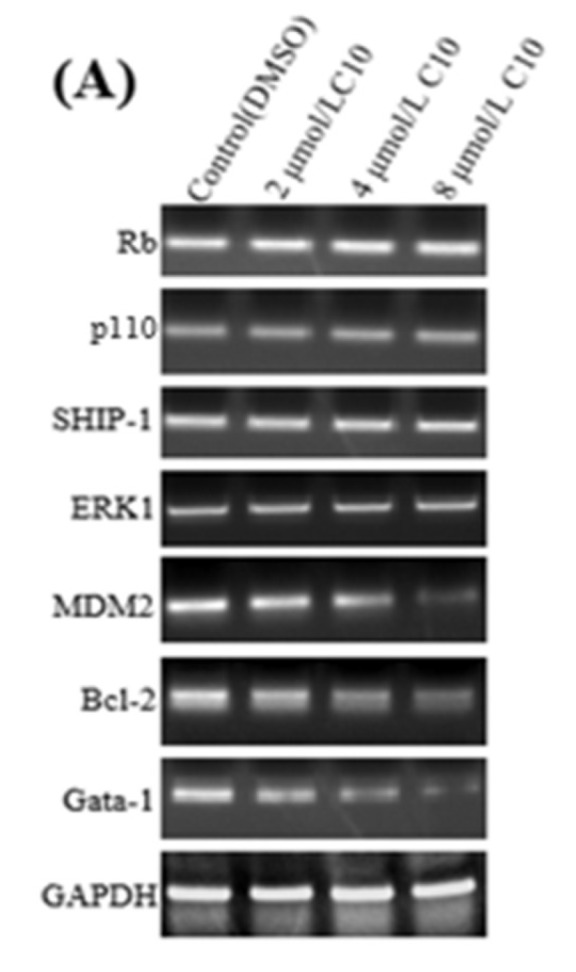
Effects of **C10** on the expression of Fli-1 target genes in PC3 cells. The regulatory effect of **C10** on expression of Fli-1 target genes at the transcription level was evaluated by RT-PCR (**A**,**B**) and qPCR (**C**), and at a protein level by Western blotting (**D**) and a quantitative analysis with Image J software. (**E**) PC3 cells were incubated with different concentrations of **C10** (0, 2, 4, and 8 μmol/L) for 24 h and then, the mRNA and protein contents were isolated. GAPDH was used as a loading control. (**F**) The schematic illustration of Fli-1 regulation of target genes governing apoptosis and proliferation. The data are represented as mean ± SD from at least three independent experiments. * *p* < 0.05 (*n* = 3) compared with the control, ** *p* < 0.01 (*n* = 3) compared with the control.

**Figure 7 ijms-21-02216-f007:**
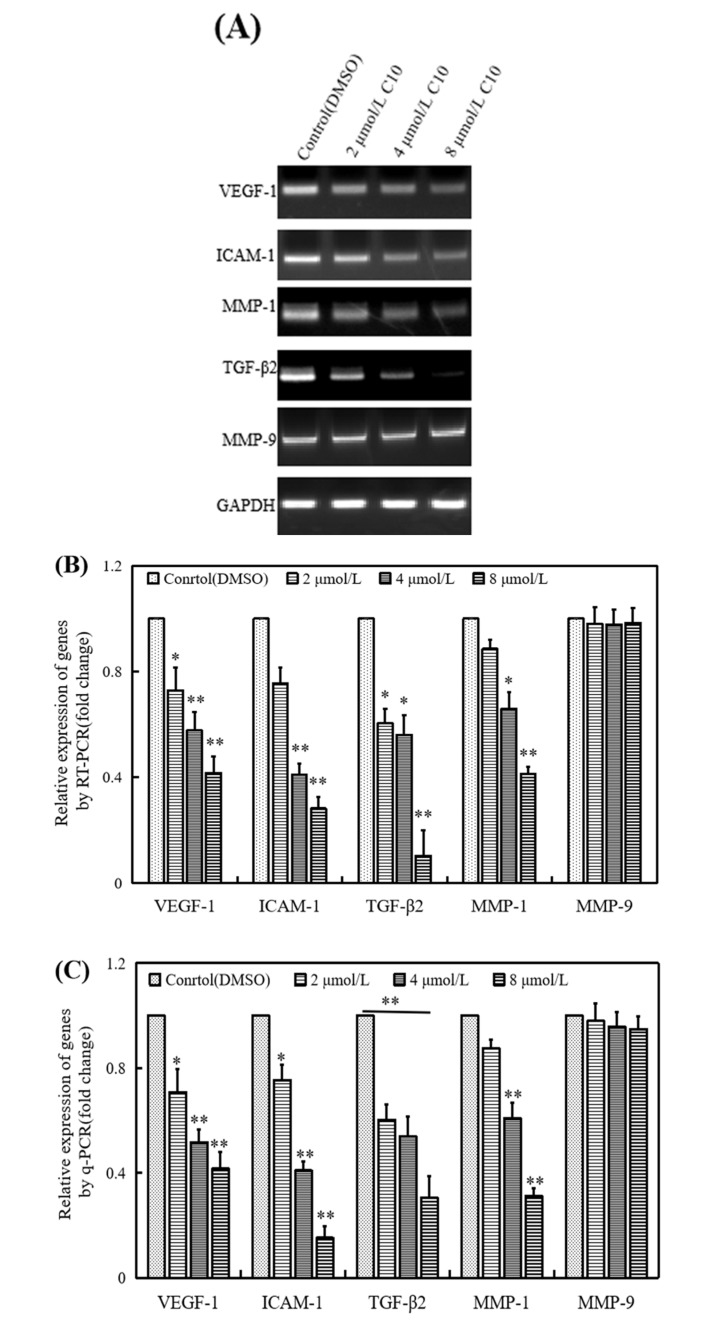
Effects of **C10** on the expression of apoptosis- and metastasis-related factors in PC3 cells. (**A**,**B**) RT-PCR, (C) qPCR, (**D**) Western blotting, and (**E**) quantitative analysis with Image J software. PC3 cells were incubated with different concentrations of **C10** (0, 2, 4, 8 μmol/L) for 24 h. (**F**) The visualization of predicted interaction of the regulatory network of Fli-1 target genes and metastasis-associated genes in PC3 cells. The network was generated with the website https://string-db.org/, using line thickness as the strength of data support and color as the type of interaction. GAPDH was used as a loading control. Experiments were performed in triplicates. The data are presented as the mean ± SEM. * *p* < 0.05 (*n* = 3) compared with the control, ** *p* < 0.01 (*n* = 3) compared with the control.

**Figure 8 ijms-21-02216-f008:**
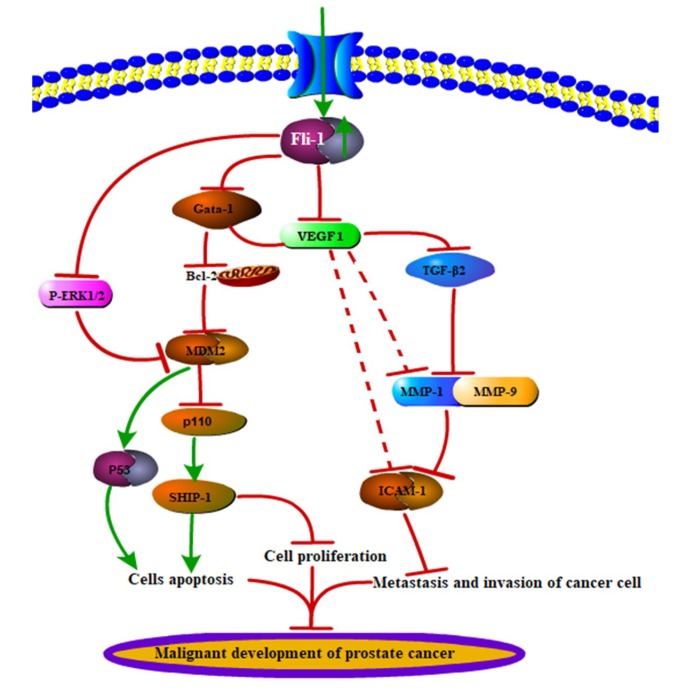
Proposed signaling pathway indicating that **C10** induces apoptosis and inhibits proliferation, metastasis, and invasion in PC3 cells. Red line: inhibition; Green line: activation; the red dotted line indicates an uncertain relationship.

**Table 1 ijms-21-02216-t001:** Primer sequences used in this study.

Gene	Primer	Primer Sequence (5′→3′)
Rb	Forward	CTCTCGTCAGGCTTGAGTTTG
	Reverse	GACATCTCATCTAGGTCAACTGC
MDM-2	Forward	GAATCATCGGACTCAGGTACATC
	Reverse	TCTGTCTCACTAATTGCTCTCCT
Bcl-2	Forward	GGTGGGGTCATGTGTGTGG
	Reverse	CGGTTCAGGTACTCAGTCATCC
Gata-1	Forward	CTGTCCCCAATAGTGCTTATGG
	Reverse	GAATAGGCTGCTGAATTGAGGG
SHIP-1	Forward	GCGTGCTGTATCGGAATTGC
	Reverse	TGGTGAAGAACCTCATGGAGAC
p110	Forward	TATTTGGACTTTGCGACAAGACT
	Reverse	TCGAACGTACGGTCTGGATAG
ERK1	Forward	TACACCAACCTCTCGTACATCG
	Reverse	CATGTCTGAAGCGCAGTAAGATT
VEGF-1	Forward	AGGGCAGAATCATCACGAAGT
	Reverse	AGGGTCTCGATTGGATGGCA
ICAM-1	Forward	ATGCCCAGACATCTGTGTCC
	Reverse	GGGGTCTCTATGCCCAACAA
TGF-β2	Forward	CAGCACACTCGATATGGACCA
	Reverse	CCTCGGGCTCAGGATAGTCT
MMP-1	Forward	AAAATTACACGCCAGATTTGCC
	Reverse	GGTGTGACATTACTCCAGAGTTG
MMP-9	Forward	TGTACCGCTATGGTTACACTCG
	Reverse	GGCAGGGACAGTTGCTTCT
Fli-1	Forward	CAGCCCCACAAGATCAACCC
	Reverse	CACCGGAGACTCCCTGGAT
GAPDH	Forward	GGAGCGAGATCCCTCCAAAAT
	Reverse	GGCTGTTGTCATACTTCTCATGG

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
