# Peer review of "Fli-1 Activation through Targeted Promoter Activity Regulation Using a Novel 3’, 5’-diprenylated Chalcone Inhibits Growth and Metastasis of Prostate Cancer Cells"

_ijms, 2020, doi:10.3390/ijms21062216_

Round 1

Reviewer 1 Report

The MS has improved, and the authors solved most issues with the previous version of the MS.

However, despite the statement that the legend to figure 3 was corrected, it is still not OK and needs correction. Especially: legend for panel B mentions: binding parameters predicted by SwissDock, but no parameters are shown in that panel, only a ribbon representation of the protein with the compound. This is the third time I make this remark, and it is still not OK.

Author Response

The MS has improved, and the authors solved most issues with the previous version of the MS. However, despite the statement that the legend to figure 3 was corrected, it is still not OK and needs correction. Especially: legend for panel B mentions: binding parameters predicted by SwissDock, but no parameters are shown in that panel, only a ribbon representation of the protein with the compound. This is the third time I make this remark, and it is still not OK.

Response: Thank you very much for your criticism and correction. We are very sorry for the mistake caused by our negligence. Since there is a certain repetition between figure 3B and figure 3C, we delete figure 3B.

Reviewer 2 Report

Line 135-136: "These results suggested that C10 directly competes with DNA to bind to Fli-1 for complex formation."

What is the docking energy for a say model DNA fragment to the binding pocket of the protein? Is it more than the ligand or less?

Line 445 "4.10. Molecular docking analysis" Please added parameters used for docking runs.

Figure 3C should be replaced with a better quality picture.

Author Response

Line 135-136: "These results suggested that C10 directly competes with DNA to bind to Fli-1 for complex formation." What is the docking energy for a say model DNA fragment to the binding pocket of the protein? Is it more than the ligand or less?

Response: Thank you very much for the question, we checked the DNA sequence of specific binding of the Fli-1, but there were no reported molecular docking between DNA and Fli-1, we also try to use the same method for investigating binding of Fli-1 and DNA, but the docking results showed that deltaG and fullfitness score had a strong correlation with DNA configuration, but the configuration of specific binding DNA sequence could not be determined, so the binding parameters could not be determined. Therefore, we changed our conclusion, we can only conclude that C10 could directly bind with Fli-1 protein for complex formation. Line 135-136 of the article was modified accordingly.

Line 445 "4.10. Molecular docking analysis" Please added parameters used for docking runs.

Response:We have added the running parameters of molecular docking as required in this paper. The parameters used for docking runs: number of BMs (binding mode) generated by the DSS engine: 5000, SD (Steepest descent) minimization steps: 100, ABNR (the adopted basis Newton-Raphson method) minimization steps: 250, number of BMs clustered and evaluated by the FullFitness:250.

Figure 3C should be replaced with a better quality picture.

Response:We have replaced a better quality pictures in the article as required.

Round 2

Reviewer 1 Report

The article is acceptable in its present form. 

This manuscript is a resubmission of an earlier submission. The following is a list of the peer review reports and author responses from that submission.

Round 1

Reviewer 1 Report

In the manuscript Ma et al., the authors examined the effect of a novel chalcone derivative on prostate cancer cell growth and invasion/migratory abilities using PC-3 prostate cancer cell lines model. A number of experiments have been performed however a major concern is that the authors have restricted all experiments to one cell line i.e. PC-3. Given that prostate cancer is a very heterogenous disease the use of a single cell line defeats the purpose of the entire study. Authors therefore should repeat at least the major findings using another cell line such as LNCaP or its androgen-independent derivative C4-2. There are several other concerns that need to be addressed by the authors (see below)

1. The authors should provide more description of Fli-1 in the introduction section, as it stands, it is difficult for a non-specialist reader to understand the subject matter.

2. The interaction between chalcone and Fli-1 requires elaboration throughout the text.

3. References should be provided for studies mentioned herein.

4. It is unclear how the data shown in figure 1Aa was normalised. Moreover, the data should also be normalised to the empty vector control as it shows as increased reporter activity on chalcone treatment. Further, it is also unclear why TERN and GLR1R were used as controls (fig 1B).

5. Figure 1C is in high discordance with 1A as it shows no changes in Fli1 protein levels upon chalcone treatment in a dose dependent manner. These data are in contrast with figure 1A that shows a dose-dependent increase in Fli-1 reporter activity.

6. While the authors have shown the induction of apoptosis upon chalcone treatment, it is unclear whether these effects are caused by chalcone-mediated Fli-1 upregulation. Authors should therefore deplete chalcone-induced Fli-1 expression using siRNA and then test whether apoptosis is induced by chalcone in order to find the mechanism of its action. Authors also should discuss whether the apoptosis is caspase dependent.

7. For figure 3H, no figure legend is provided.

8. Figure 4A: Authors should rule out the possibility that the decrease in invasion was not caused by decreased cell number caused by C10 toxicity. To do this they should use doses of C10 that are non toxic during the time frame studied.

9. Migration experiment figure 4C: Authors should employ a complete wound closure as control and use non-lethal doses and time points to rule out the possibility that the decrease in migration was not a result of decreased cell number.

10. Figure 5A: A densitometry analysis of protein bands should be performed.

11. Figure 5A: the rationale of showing the decrease in MDM2 levels by C10 appears to be unclear. Authors should test the levels of MDM2 target p53 to evaluate the effect od C10 mediated MDM2 decrease on p53 levels using a Western blot assay.

12. Figure 6A: MMP-1 and 9 expression levels doesn’t necessarily correlate with proteolytic activity, so this should be measured by zymography.

13. Discussion: FLI-1 linking to breast cancer should be in introduction

14. Formatting issues: C10 sometimes in bold sometimes not, more consistency needed

Author Response

In the manuscript Ma et al., the authors examined the effect of a novel chalcone derivative on prostate cancer cell growth and invasion/migratory abilities using PC-3 prostate cancer cell lines model. A number of experiments have been performed however a major concern is that the authors have restricted all experiments to one cell line i.e. PC-3. Given that prostate cancer is a very heterogenous disease the use of a single cell line defeats the purpose of the entire study. Authors, therefore, should repeat at least the major findings using another cell line such as LNCaP or its androgen-independent derivative C4-2. There are several other concerns that need to be addressed by the authors (see below)

1. The authors should provide more description of Fli-1 in the introduction section, as it stands, it is difficult for a non-specialist reader to understand the subject matter. The interaction between chalcone and Fli-1 requires elaboration throughout the text.

Response: Thanks for the reviewer’s suggestions, we have added information about interaction between chalcone and Fli-1 in this article.

2. References should be provided for studies mentioned herein.

Response: We have changed in the manuscript.

3. It is unclear how the data shown in figure 1A was normalised. Moreover, the data should also be normalised to the empty vector control as it shows as increased reporter activity on chalcone treatment. Further, it is also unclear why TERN and GLR1R were used as controls (Fig 1B).

Response:We referred to the presentation of reference 23 (Liu et al., 2017) and took luciferase activity as the vertical coordinate, which indicated that Luciferase activity reflects the promoter activity. The promoter sequence of TERN and GLR1R was not specific with Fli-1, so TERN and GLR1R were used as the non-specific negative control.

4. Figure 1C is in high discordance with 1A as it shows no changes in Fli1 protein levels upon chalcone treatment in a dose dependent manner. These data are in contrast with figure 1A that shows a dose-dependent increase in Fli-1 reporter activity.

Response: Indeed, there was a dose-dependent increase in Fli-1 reporter activity in figure 1A, while the protein test found no dose-dependent increase. The reasons for this difference are under further study, but relevant data are not yet available.

5. While the authors have shown the induction of apoptosis upon chalcone treatment, it is unclear whether these effects are caused by chalcone-mediated Fli-1 upregulation. Authors should therefore deplete chalcone-induced Fli-1 expression using siRNA and then test whether apoptosis is induced by chalcone in order to find the mechanism of its action. Authors also should discuss whether the apoptosis is caspase dependent.

Response: Thank you for your good suggestions. We have added related results in the manuscript.

6. For figure 3H, no figure legend is provided.

Response: Thank you for your reminder. We have deleted the figure 3G and H in the manuscript.

7. Figure 4A: Authors should rule out the possibility that the decrease in invasion was not caused by decreased cell number caused by C10 toxicity. To do this they should use doses of C10 that are non toxic during the time frame studied.

Response: Thank you for your suggestions. We have considered this issue before doing the experiment. According to the experimental results of flow cytometer assay, 0.5, 1 and 2 μmol/L of C10 were almost non-toxicity to PC3 cells; the state of PC3 cells did not change and did not cause apoptosis. Therefore, we have checked the effects of lower doses of C10 (0.5, 1 and 2 μmol/L) on the invasion of PC3 cells.

8. Migration experiment figure 4C: Authors should employ a complete wound closure as control and use non-lethal doses and time points to rule out the possibility that the decrease in migration was not a result of decreased cell number.

Response: Thank you for your suggestions. We have considered this issue before doing the experiment. According to the experimental results of flow cytometer assay, 0.5, 1 and 2 μmol/L of C10 were almost non-toxicity to PC3 cells; the state of PC3 cells did not change and did not cause apoptosis. Therefore, we have checked the effects of lower doses of C10 (0.5, 1 and 2 μmol/L) on the migration of PC3 cells.

9. Figure 5A: A densitometry analysis of protein bands should be performed.

Response: We appreciate your reminder. We have added densitometry analysis in the manuscript.

10. Figure 5A: the rationale of showing the decrease in MDM2 levels by C10 appears to be unclear. Authors should test the levels of MDM2 target p53 to evaluate the effect od C10 mediated MDM2 decrease on p53 levels using a Western blot assay.

Response: Thank you for your suggestions. In the manuscript, we have supplemented the protein expression level of P53.

11. Figure 6A: MMP-1 and 9 expression levels doesn’t necessarily correlate with proteolytic activity, so this should be measured by zymography.

Response: Thank you for your suggestions. However, due to the lack of relevant materials for zymography, the relevant data of zymography are not supplemented in this paper. We will supplement the detection results of zymography in the ongoing research.

12. Discussion: Fli-1 linking to breast cancer should be in introduction

Response: Thank you for your suggestions. We have provide the information of Fli-1 linking to breast cancer in the manuscript. 13.Formatting issues: C10 sometimes in bold sometimes not, more consistency needed Response: Thank you for your reminder. We have changed in the manuscript.

Reviewer 2 Report

Very major remark:

It is striking that the compound C10 is presented here as an activator of Fli-1, and to induce strong upregulation of Fli-1 (Fig. 1C), while the compound was first described by the group to inhibit Fli-1 expression and proposed as an Fli-1 inhibitor in other cell types (reference 18 in the submitted MS, Zhang et al.).  This aspect and the apparent discrepancy is not referred to in the MS.

Similarly, the introduction contains the following statement: “the effect of C10 on prostate cancer cell growth …. has not been elucidated.” The group actually studied C10 on PC3 cells in cell growth assays, apoptosis and cell cycle progression in their reference 15 article (Wen et al., European journal of medicinal chemistry, 2017). The whole section 2.2 is therefore redundant, while reference 15 is not referred to in section 2.2. Strikingly, the conclusions and results of effects of C10 on cell cycle progression in this paper and the currently submitted article are different.

Similarly, the same group has previously found other inhibitors (A661,A665) of Fli-1 and demonstrated their inhibitory effect on Fli-1, and also demonstrated that these compounds inhibit PC3 cell growth (Liu et al., Cell Death Dis. 2019 Feb; 10(2): 117) . This suggests that inhibition of Fli-1 in PC3 inhibits PC3 cell growth.

Similarly, the authors reviewed the role of Fli-1 in cancer in reference 25 in the submitted MS (Liu et al,.Oncogene, 2015). In this review, it is also clearly stated that Fli-1 inhibitors, rather than activators are developed for treatment of different cancer types. Also, Fli1 expression leads to overexpression of Bcl-2,VEGF and Mdm-2 and to activation of a PI3K/Ras pathway resulting in Erk phosphorylation. In contrast, in the current MS, the authors claim that C10 is an Fli-1 activator that reduces expression of Bcl-2, VEGF and Mdm-2 and Erk phosphorylation. Figure 7 in the submitted MS is in many aspects the opposite of figure 1 in the Liu et al. review. These striking unexpected differences are not mentioned at all. In this MS, the authors also state: “Thus, putative inhibitors should be analyzed side-by-side with specific shRNA/RNAi for Fli-1 to determine if they phenocopy the effect of knocking down this TF.” This would indeed be necessary in the PC3 cell line to check whether the proposed mechanism makes sense here.

To reconcile the probably unexpected finding that C10 is an activator of Fli-1, associated with effects on target genes that one might expect with inhibition of Fli-1, the authors fully focus on reference 28. Ref 28 is in fact wrong in the reference list (it references Song et al.), and probably should refer to Scheiber et al.  (Neoplasia. 2014 Oct 23;16(10):801-13.). However, citing Yan et al., (Cancer Med. 2018 Aug; 7(8): 3548–3560.), “the role of FLI‐1 in breast cancer is still controversial. The mainstream studies, including our previous study, suggest that FLI‐1 inhibits cell apoptosis or promotes tumor progression in breast cancer.13, 14, 15 However, Scheiber et al. presented a contradictory conclusion that reduced expression of FLI‐1 promotes tumor progression.” , using Scheiber et al. as sole base to support the findings in the submitted MS. also requires a lot of nuance and proper literature context (which is again lacking).

It seems that the authors are very, very selective, and very biased in their citation of literature and it is hard to believe that the authors have forgotten their own results which for the majority of data seems to be at odds with the data presented here, or even already produced a complete section recycled in the current submission. It seems that the MS deliberately omits mentioning facts and data that do not fit the results obtained in this article. I therefore strongly advice not to accept this MS for publication.

Other remarks:

Putative inhibitors should be analyzed side-by-side with specific shRNA/RNAi for Fli-1 to determine if they phenocopy the effect of knocking down this TF

No proper controls are shown for the thermal shift assay. As the compound seems to induce expression of Fli-1 in PC3 cells, it may be that the increase in Western blot reflects increased expression due to the compound. It will be necessary to also show the intensities of the bands for the different compound concentrations without the thermal denaturation.

For the western blots, only cut out strips of the selected bands are shown. It is also necessary to show the original gels with the MW markers (e.g. in supplement)

The authors show a western blot and PCR data for a gene/protein called PI3-K/Ras: ????????????

What is that? Based on the choice of antibody mentioned in the materials and methods, it seems to be p110, a PI3-K subunit, and Ras is untested here.

Comparing morphologies based on the tiny cell pictures in Fig 3, panel B and C is impossible.

When mentioning the thermal shift assay in 2.1, the refernce is missing, the MS  mentions: “(add citation…)”

Fli-1 is totally missing in the introduction.

Figure 7 is strange, many of the inhibitory arrows ahould be activating arrows?: e.g. VEGF does not inhibit MDM2 or PI3K or Ras

Figure 2 contains an error: the depicted panel C should be panel B, while the panel C mentioned in the legend is missing.

The “semi-quantitative qPCR” images in the article figures are not qPCR, but regular PCR data?

Author Response

It is striking that the compound C10 is presented here as an activator of Fli-1, and to induce strong upregulation of Fli-1 (Fig. 1C), while the compound was first described by the group to inhibit Fli-1 expression and proposed as an Fli-1 inhibitor in other cell types (reference 18 in the submitted MS, Zhang et al.). This aspect and the apparent discrepancy is not referred to in the MS.

Response: Indeed, we also found that C10 can significantly inhibit Fli-1 expression in leukemia cells HEL (this study is still in further systematic studies and has not been fully confirmed). However, we truly found that this compound can significantly up-regulate the expression of Fli-1 protein in prostate cancer cells PC3. These two conclusions are not contradictory because the two types of cancer cells are fundamentally different, but the reason for this diametrically opposite result is that we don't know yet. We are continuing the subject of systematic research and believe that we will find the cause of this result. In addition, we briefly introduced those resullts in introduction of the article and gave the necessary explanations.

Similarly, the introduction contains the following statement: “the effect of C10 on prostate cancer cell growth …. has not been elucidated.” The group actually studied C10 on PC3 cells in cell growth assays, apoptosis and cell cycle progression in their reference 15 article (Wen et al., European journal of medicinal chemistry, 2017). The whole section 2.2 is therefore redundant, while reference 15 is not referred to in section 2.2. Strikingly, the conclusions and results of effects of C10 on cell cycle progression in this paper and the currently submitted article are different.

Response:We did a preliminary study of the growth assay, apoptosis, and cell cycle progression of PC3 cells by C10 in our previous published paper (Reference 18). However, this article is a research paper focusing on chemical synthesis of 3',5'-diprenylated chalcone derivatives and their primary cytotoxicity. Only the inhibition of 5 μM / L of C10 on cell growth, apoptosis, and cell cycle has been tested in the cell viability section. No concentration gradient treatment was performed. On the basis of this study, the effects of different treatment time and different concentrations of C10 on cell growth were studied in different prostate cancer cells. This study is a more in-depth and systematic study. Therefore, it is not that the entire 2.2 section is redundant, but rather necessary. The results of the C10 study of the cell cycle did appear different, and we are now studying the reasons for the different results. In view of this, the results of the cell cycle research with C10 have little effect on the discussion of the mechanism part of this article, so we decided to eliminate this result after discussion. Similarly, the same group has previously found other inhibitors (A661,A665) of Fli-1 and demonstrated their inhibitory effect on Fli-1, and also demonstrated that these compounds inhibit PC3 cell growth (Liu et al., Cell Death Dis. 2019 Feb; 10(2): 117) . This suggests that inhibition of Fli-1 in PC3 inhibits PC3 cell growth.

Similarly, the authors reviewed the role of Fli-1 in cancer in reference 25 in the submitted MS (Liet al,.Oncogene, 2015). In this review, it is also clearly stated that Fli-1 inhibitors, rather than activators are developed for treatment of different cancer types. Also, Fli1 expression leads to overexpression of Bcl-2,VEGF and Mdm-2 and to activation of a PI3K/Ras pathway resulting in Erk phosphorylation. In contrast, in the current MS, the authors claim that C10 is an Fli-1 activator that reduces expression of Bcl-2, VEGF and Mdm-2 and Erk phosphorylation. Figure 7 in the submitted MS is in many aspects the opposite of figure 1 in the Liu et al. review. These striking unexpected differences are not mentioned at all. In this MS, the authors also state: “Thus, putative inhibitors should be analyzed side-by-side with specific shRNA/RNAi for Fli-1 to determine if they phenocopy the effect of knocking down this TF.” This would indeed be necessary in the PC3 cell line to check whether the proposed mechanism makes sense here.

Response:Our other team in the same laboratory studied that Fli-1 inhibitors (A661, A665) can indeed inhibit the growth of PC3 cells. However, it does not explain the regulation of Fli-1 agonists on the growth of prostate cancer cells, and the study did not focus on the changes of Fli-1 after these two compounds treated prostate cancer cells. In fact, consulting the literature did not find any related research on Fli-1 and prostate cancer. Therefore, we insisted on our findings in the research. It may be inconsistent with many existing reports in other cancer cells, but we still believe in the innovation and exploratory of these research results. For review article published previously by our group (Literature 31), the role of Fli-1 in cancer is reviewed (Li et al., Oncogene, 2015), and it is clear that Fli-1 inhibitors, not activators, are used to treat different types of cancer. This review does focus only on Fli-1 inhibitors and hematological tumors, and so far there have been no reports on the role of Fli-1 agonists in prostate cancer cells. It was indeed found herein that C10 is a Fli-1 activator that can reduce the expression of Bcl-2, VEGF, Mdm-2 and Erk phosphorylation. This is inconsistent with the conclusion that Fli-1 expression discussed in the review article leads to overexpression of Bcl-2, VEGF, and Mdm-2, and activates the PI3K / Ras pathway, leading to Erk phosphorylation. However, this does not mean that the research conclusions in this article are incorrect, because the review article focuses on the role of Fli-1 in hematological tumors, rather than related research reports on Fli-1 agonists and prostate cancer. There are some reasons for the differences between the two types of conclusion, but the reason we don't know until now. There are indeed many possible reasons involved, and we dare not infer them. Therefore, these surprising and unexpected differences are not mentioned at all in this study, and this article is only concerned with the findings of this article and does not want to get involved in this debate. The research group is also further in-depth systematic research into the internal causes of "these amazing unexpected differences", I believe there will be new discoveries. In addition, thanks the reviewer’s suggestions for “analysing Fli1 with specific shRNA / RNAi to determine whether they have the effect of knocking out this TF”, which is indeed necessary in PC3 cell lines to check does the mechanism make sense here. Therefore, we used siRNA to knock down Fli-1 expression in C10-induced Fli-1 overexpressing PC3 cells, and then examined the recovery of cell growth ability, which indicates that C10 as a Fli-1 agonist induces Fli-1 over-expression can indeed inhibit the growth of prostate cancer cells.

To reconcile the probably unexpected finding that C10 is an activator of Fli-1, associated with effects on target genes that one might expect with inhibition of Fli-1, the authors fully focus on reference 28. Ref 28 is in fact wrong in the reference list (it references Song et al.), and probably should refer to Scheiber et al. (Neoplasia. 2014 Oct 23;16(10):801-13.). However, citing Yan et al., (Cancer Med. 2018 Aug; 7(8): 3548–3560.), “the role of FLI‐1 in breast cancer is still controversial. The mainstream studies, including our previous study, suggest that FLI‐1 inhibits cell apoptosis or promotes tumor progression in breast cancer.13, 14, 15 However, Scheiber et al. presented a contradictory conclusion that reduced expression of FLI‐1 promotes tumor progression.” , using Scheiber et al. as sole base to support the findings in the submitted MS. also requires a lot of nuance and proper literature context (which is again lacking).

Response: We are very grateful for reviewer’s suggestions for the inaccuracies and inadequacies of the literature citations. We have revised it in the manuscript and added relevant information in the discussion section. It seems that the authors are very, very selective, and very biased in their citation of literature and it is hard to believe that the authors have forgotten their own results which for the majority of data seems to be at odds with the data presented here, or even already produced a complete section recycled in the current submission. It seems that the MS deliberately omits mentioning facts and data that do not fit the results obtained in this article. I therefore strongly advice not to accept this MS for publication. Response: It is clear that the results of this paper are inconsistent with the results of most existing studies, and this paper only reports the results of this study in a real way. As for the deficiency of literature citation, it is really our negligence. When we wrote this article, we did not quote the literature without prejudice, but we quoted relevant literature according to our needs. Thank you very much for the valuable suggestions from the reviewers and give us an opportunity to further study the issues involved in this topic.

Other remarks:

Putative inhibitors should be analyzed side-by-side with specific shRNA/RNAi for Fli-1 to determine if they phenocopy the effect of knocking down this TF

Response:We have added relevant research content, specifically using siRNA to knock down Fli-1 expression in PC3 cells with low concentration (cells are not killed and induce apoptosis) C10-induced Fli-1 overexpression and then detect recovery of cell growth capacity. The results showed that after reducing the expression of Fli-1, the growth ability of PC3 cells induced by low concentration of C10 induced Fli-1 overexpression was significantly restored. This indicates that C10, as a Fli-1 agonist, can indeed inhibit the growth of prostate cancer cells by inducing Fli-1 overexpression.

No proper controls are shown for the thermal shift assay. As the compound seems to induce expression of Fli-1 in PC3 cells, it may be that the increase in Western blot reflects increased expression due to the compound. It will be necessary to also show the intensities of the bands for the different compound concentrations without the thermal denaturation.

Response:We have added the protein expression assays of Fli-1 expression at different concentrations of compound treatment without thermal denaturation.

For the western blots, only cut out strips of the selected bands are shown. It is also necessary to show the original gels with the MW markers (e.g. in supplement)

Response:We have provided all original photos of WB including MW markers as supplementary materials at the submission.

The authors show a western blot and PCR data for a gene/protein called PI3-K/Ras: ????????????

Response: Thank you for your reminder. We have made corrections in the manuscript.

What is that? Based on the choice of antibody mentioned in the materials and methods, it seems to be p110, a PI3-K subunit, and Ras is untested here.

Response: Thank you for your valuable reminder. We have made corrections in the manuscript.

Comparing morphologies based on the tiny cell pictures in Fig 4, panel B and C is impossible.

Response:Figure 4B is the picture taken in the bright field of fluorescence inverted microscope, figure 4C is the original picture of the C10-treated cells staining with Hoechst 33258 using fluorescence inverted microscope. All pictures were the original picture without any modification.

When mentioning the thermal shift assay in 2.1, the refernce is missing, the MS mentions: “(add citation…)”

Response: Thank you for your suggestions. We have added this reference in the manuscript.

Fli-1 is totally missing in the introduction.

Response: Thanks to the reviewer’s suggestions, we have added necessary information of Fli-1 in the introduction.

Figure 7 is strange, many of the inhibitory arrows ahould be activating arrows?: e.g. VEGF does not inhibit MDM2 or PI3K or Ras

Response: We have made corrections in the manuscript.

Figure 2 contains an error: the depicted panel C should be panel B, while the panel C mentioned in the legend is missing.

Response: We have made corrections in the manuscript.

The “semi-quantitative qPCR” images in the article figures are not qPCR, but regular PCR data?

Response: Thank you very much for your reminder, we have corrected it in the manuscript.

Round 2

Reviewer 1 Report

na

Reviewer 2 Report

Dear Editor,

I have re-read the revised MS and still feel that the article has significant shortcomings and I feel it does not match the standards of IJMS.

To begin with, I very much dislike the fact that the authors had cherry-picked literature in their original submission, omitting in fact most of the literature on Fli-1 inhibitors, including and especially their own work, and that the authors selectively cited an article that was in line with their data, omitting the vast majority of literature that was not in line with the data.
I have no problem at all with work in a certain cell type that indicate clear differences with a lot of existing literature in other cell types, as this is perfectly possible. However, I find it very bad scientific conduct to
simply omit references that may raise questions or doubts (e.g. of the reviewers). The authors state in their rebuttal: "When we wrote this article, we did not quote the literature without prejudice, but we quoted relevant literature to our needs".

However, even after the highly critical comments that "a lot of literature that did not seem to fit the current findings" was omitted, the authors chose not to mention potentially conflicting data in the revised version. For example, despite critical comments on highly selective literature citation, the authors still do not mention at all that they previously discovered C10 as a Fli-1 antagonist inhibiting Fli-1 expression in other cell lines, while C10 is presented here as an agonist increasing Fli-1 expression in prostate cancer cells.
The rebuttal claim that the authors now briefly introduced those previous results in the introduction is misleading, as they do not mention that all effects in leukemia cells are in fact the opposite of what is found in PC3 cells.

The effects of C10 as Fli-1 agonist on pathways is in many aspects completely the opposite as what is found in haematological cancers. However, again, this is still not discussed or even mentioned.
The authors agree in their rebuttal that the effect of C10 on Fli-1 and the resulting effects in PC3 cells oppose what is found in haematological cancers, but state that "...the reason we don't know until now. There are indeed many possible reasons involved and we dare not infer them.
Therefore these surprising and unexpected differences are not mentioned at all in this study, and this article is only concerned with the findings of this article and does not want to get involved in this debate."
If we follow along this line, we should make scientific articles completely descriptive, abolish the discussion sections of scientific articles and leave all discussion for review articles.

Similarly, the authors previously already performed quite some characterization of C10 in PC3 cells which is still not referred to in the revised version of sections 2.2 and 2.3, where any characterization is suggested as completely new. The rebuttal that a concentration gradient had not been performed in the previous work does not mean that the previous work should be left unmentioned.
Moreover, the rebuttal that no concentration gradient of C10 was tested on growth of PC3 cells is simply false, as a full concentration gradient of C10 in PC3 cells was presented in the previous article.

As requested, the authors now delivered the original western blots in supplement, as only selected strips are shown in the MS.
In some cases none of the presented western blots in addendum match the western blot shown in the figures. In contrast to my request, in most cases, not the full western blot is shown, but again only strips are shown of the original blot in supplement and it is unclear why the authors chose to only show parts of the original blots. However, the blots of figure 1C and 1D that do show larger portions of the blot seem to indicate that the GAPDH controls were run on other gels than the subject proteins under investigation, which is of course not the way to investigate differences in protein expression via western blot, as there is no control at all over intentional or non-intentional loading artifacts.

Figure 2C is totally unclear to me, as the X axis does not match the legend.

The panels in Figure 3 are still wrong as in the original MS, although the authors replied in their rebuttal that the figure was corrected.